# Visible 405 nm Violet-Blue Light Successfully Inactivates HIV-1 in Human Plasma

**DOI:** 10.3390/pathogens11070778

**Published:** 2022-07-08

**Authors:** Viswanath Ragupathy, Mohan Haleyurgirisetty, Neetu Dahiya, Caitlin Stewart, John Anderson, Scott MacGregor, Michelle Maclean, Indira Hewlett, Chintamani Atreya

**Affiliations:** 1Office of Blood Research and Review, Center for Biologics Evaluation and Research, Food and Drug Administration, Silver Spring, MD 20993, USA; mohan.haleyurgirisetty@fda.hhs.gov (M.H.); neetu.dahiya@fda.hhs.gov (N.D.); indira.hewlett@fda.hhs.gov (I.H.); 2The Robertson Trust Laboratory for Electronic Sterilization Technologies, Department of Electronic and Electrical Engineering, University of Strathclyde, Glasgow G1 1XQ, UK; caitlin.stewart@strath.ac.uk (C.S.); j.g.anderson@strath.ac.uk (J.A.); scott.macgregor@strath.ac.uk (S.M.); michelle.maclean@strath.ac.uk (M.M.); 3Department of Biomedical Engineering, University of Strathclyde, Glasgow G1 1XQ, UK

**Keywords:** PRT, blue light, pathogen inactivation, HIV-1, virucidal, 405 nm

## Abstract

Despite significant advances in ensuring the safety of the blood supply, there is continued risk of transfusion transmitted infections (TTIs) from newly emerging or re-emerging infections. Globally, several pathogen reduction technologies (PRTs) for blood safety have been in development as an alternative to traditional treatment methods. Despite broad spectrum antimicrobial efficacy, some of the approved ultraviolet (UV) light-based PRTs, understandably due to UV light-associated toxicities, fall short in preserving the full functional spectrum of the treated blood components. As a safer alternative to the UV-based microbicidal technologies, investigations into the use of violet-blue light in the region of 405 nm have been on the rise as these wavelengths do not impair the treated product at doses that demonstrate microbicidal activity. Recently, we have demonstrated that a 405 nm violet-blue light dose of 270 J/cm^2^ was sufficient for reducing bacteria and the parasite in plasma and platelets suspended in plasma while preserving the quality of the treated blood product stored for transfusion. Drawn from the previous experience, here we evaluated the virucidal potential of 405 nm violet-blue light dose of 270 J/cm^2^ on an important blood-borne enveloped virus, the human immunodeficiency virus 1 (HIV-1), in human plasma. Both test plasma (HIV-1 spiked and treated with various doses of 405 nm light) and control plasma (HIV-1 spiked, but not treated with the light) samples were cultured with HIV-1 permissive H9 cell line for up to 21 days to estimate the viral titers. Quantitative HIV-1 p24 antigen (HIV-1 p24) levels reflective of HIV-1 titers were measured for each light dose to assess virus infectivity. Our results demonstrate that a 405 nm light dose of 270 J/cm^2^ is also capable of 4–5 log HIV-1 reduction in plasma under the conditions tested. Overall, this study provides the first proof-of-concept that 405 nm violet-blue light successfully inactivates HIV-1 present in human plasma, thereby demonstrating its potential towards being an effective PRT for this blood component safety.

## 1. Introduction

Threats from known and unknown transfusion-transmitted infections (TTIs) are a major public health safety concern for ex vivo stored blood components (plasma, packed red blood cells, and platelet concentrates), and the field is cautiously advancing towards developing robust pathogen inactivation (PI) treatments, also known as Pathogen Reduction Technologies (PRTs), for whole blood safety [1,2,3]. The concept of PRT for blood safety has origins which coincide with the discovery of HIV and its severe impact on the safety of blood transfusions. Although the risk of transmission of infections via blood is relatively very low due to the implementation of several layers of safety, residual risk is still unavoidable. It is well established that the PI/PRTs being used or developed around the world, which are either currently approved in some countries or in the experimental stage, have demonstrated unintended consequences on the quality and efficacy of transfusion products, whilst being effective in reducing either pathogen levels or their infectivity. Therefore, the field has not yet generated an effective technology that balances both these essential parameters, and hence this continues to be an unmet need in transfusion medicine [4,5]. Our research focus has been in identifying and evaluating new approaches to pathogen reduction in stored human plasma and in room temperature stored platelet concentrates that are safer relative to the existing solvent detergent or ultraviolet (UV) light-based technologies and which show promise to be more selective in pathogen inactivation, whilst sparing the full functional spectrum of the product.

To date, we have utilized high intensity narrow spectrum (HINS) 405 nm violet-blue light that falls within the visible light spectrum and successfully demonstrated its potential as a pathogen reduction tool for human plasma and platelets stored in plasma, by experimentally contaminating the two blood components with a number of bacteria as well as a blood-borne protozoan parasite (*Trypanosoma cruzi*) and feline calicivirus (FCV) [6,7,8,9,10]. We have also demonstrated that the violet-blue visible light-treated platelets behave similar to the light-untreated platelets in vivo in a severe combined immunodeficient (SCID) mouse model, and that there is no visible changes to protein integrity or detection of protein oxidation in human plasma treated with 405 nm light doses that completely inactivate bacteria and protozoan parasites [8,9,10]. 

In this report, 405 nm light was evaluated on human plasma spiked with high concentrations of HIV-1 to determine whether this enveloped virus can be inactivated in plasma. The results demonstrate that inactivation of HIV-1 in human plasma is successful with violet-blue light treatment. This observation, together with our previously demonstrated proof-of-concept studies on the pathogen inactivation potential of 405 nm light, lends credence to the vision that 405 nm light can be harnessed as a broad-spectrum pathogen reduction technology for further evaluation towards blood safety. 

## 2. Materials and Methods

### 2.1. Human Plasma and HIV-1 Virus

Human platelet poor plasma (PPP) was prepared in the laboratory from six individual donations-derived platelet concentrate (PC) units (*n* = 6) obtained from the National Institutes of Health Blood Bank (Bethesda, MD, USA). The plasma was either used fresh or thawed from a frozen (−80 °C) stock. This study involving human subjects’ protocol was approved by FDA Research Involving Human Subjects Committee (RIHSC, Exemption Approval #11-036B). The HIV-1 susceptible human H9 cell line (Cat No: ARP-87, derivative of HUT 78 cell line) was obtained from the National Institutes of Health (NIH) AIDS Reagent Program (Germantown, MD, USA). The cells were cultured at 37 °C in 5% CO_2_ in RPMI 1640 medium containing 10% fetal calf serum, 2 mm glutamine, 50 µg/mL penicillin, and 50 µg/mL streptomycin. HIV-1 virus MN (MN) was originally obtained from the NIH AIDS Reagent Program and reproduced in our laboratory using H9 cells and stored as high titer stocks. The HIV MN is a lab-adapted subtype B virus isolate, which was used in several HIV-1 in vitro studies [11,12]. For the current study, HIV-1 virus stock was diluted to 10 ng/mL of HIV-1 p24 with plasma of a healthy blood donor.

### 2.2. 405 nm Violet-Blue Light Treatment of the Virus

All experiments involving 405-nm light treatment of plasma were performed using a closable prototype system (US Patent Application no. 62/236, 706, 2015), which contained a light source composed of multiple narrow band 405 nm LED arrays (FWHM ~20 nm; LED Engin, CA, USA), with appropriate thermal management and powered by LED drivers (Mean Well, New Taipei City, Taiwan). The light at the sample surface was measured using a radiant power meter and photodiode detector (LOT-Oriel Ltd., Surrey, United Kingdom). The light source in the closed system was fixed at 14 cm above a shaker platform on which the samples were placed. This system is referred to as ‘the light device’ in this report.

As illustrated in Figure 1, two 12-well culture plates, one marked as ‘Test’ and the other as ‘Control’ were used for each experiment. Each 12-well plate was marked to have replicate wells for each time point (15 min, 30 min, 1 h, 4 h and 5 h and negative plasma control). One mL of HIV-1 (10 ng/mL) spiked plasma was placed in each well except for the two negative plasma control wells having only 1 mL plasma with no virus (Figure 1). For 405 nm light treatment, the ‘Test’ plate was placed on the shaker platform of the device, set at 72 rpm, and exposed to 15 mW/cm^2^ 405 nm light for a period of up to 5 h (270 J/cm^2^) at 22 °C as previously described [9]. At indicated time intervals equivalent to a specific light dose (15 min = 13.5 J/cm^2^; 30 min, = 27 J/cm^2^; 1 h, = 54 J/cm^2^; 4 h, = 216 J/cm^2^ and 5 h, = 270 J/cm^2^), one mL plasma was transferred from the designated wells to a storage tube and immediately stored frozen (−80 °C) until viral infectivity assays were performed. The ‘Control’ plate containing HIV-1 spiked plasma was incubated in the biological safety hood (with no violet-blue light treatment), with samples collected and stored following the same protocol as for the ‘Test’ samples.

### 2.3. Estimation of HIV-1 Infectivity in Plasma Samples

To assess HIV-1 infectivity in the virus permissive cell H9 cells, two 12-well plates were seeded with H9 cells at a concentration of 1 × 10^4^ cells/well and incubated for 24 h at 37 °C, in 5% CO_2_. The H9 cell suspension from each well was then centrifuged and the cell pellet was washed in serum-free media twice and reconstituted in 0.25 mL serum-free media. Frozen 1 mL plasma from the test and control plates were thawed to room temperature prior to performing the HIV-1 infectivity assay using H9 cells. Briefly, to assess HIV-1 infectivity, H9 cells were mixed with either HIV-1 spiked, violet-blue light-treated plasma from each well of the test plate, or HIV-1 spiked untreated plasma from each well of the control plate and incubated at 37 °C, in 5% CO_2_ for 2 h. The cells were then washed twice with PBS to remove cell-free virus, and cell pellets from both test and control samples were reconstituted with RPMI medium containing 10% fetal bovine serum and cultured for up to 21 days in an incubator at 37 °C, in 5% CO_2_. Culture supernatants were collected on indicated days (0, 7, 14 and 21) and stored for HIV-1 p24 measurements reflective of the viable virus infectivity on all samples at one time. 

HIV-1 infectivity was quantitated from cell-free culture supernatant using an Alliance HIV-1 p24 Antigen ELISA Kit (Cat # NEK050B001KT, Perkin Elmer, Waltham, MA, USA) according to the manufacturer’s instructions. 

### 2.4. Statistical Analysis

HIV-1 p24 values were log transformed and presented as mean (*n* = 2) ± standard deviation. Statistical significance in viral titers were determined via the two-way ANOVA using GraphPad Prism 7.03 (GraphPad Software, Inc., San Diego, CA, USA). Statistical significance indicated by * *p* > 0.05 or *** *p* < 0.001 where applicable.

## 3. Results

### 405 nm Light Inactivates HIV-1 in Human Plasma

In vitro studies with HIV-1 spiked plasma were designed to test whether 405 nm light can inactivate HIV-1 in human plasma and to determine the optimal light dose that can achieve maximum inactivation of the virus. For this pilot experiment, 25 mL plasma of a donor randomly selected from the six donor plasma samples collected, was spiked with 1 mL HIV-1 (10 ng/mL HIV-1 p24) and exposed to doses of 405 nm light ranging from 13.5 J/cm^2^ to 270 J/cm^2^ along with respective controls (no light treatment). Virus inactivation potential of the light and subsequent residual survival of HIV-1 in plasma were monitored by evaluating the treated plasma for HIV-1 infectivity using H9 cells. The results of this experiment are illustrated in Figure 2. All plasma samples spiked with HIV-1 but not subjected to light treatment (i.e., controls for the light treatment) demonstrated a significant increase (*p* < 0.001) of virus titer from baseline of 1.4 log HIV-1 p24 pg/mL to a titer of 7 log HIV-1 p24 pg/mL by day 21 (Figure 2A). For plasma samples spiked with the virus and exposed to light doses ranging from 13.5 J/cm^2^ to 270 J/cm^2^, it was observed that while HIV-1 inactivation was variable in this donor plasma sample with light doses up to 216 J/cm^2^, there was a consistent pattern of maximum inactivation of HIV-1 from ~7 log HIV-1 p24 pg/mL down to ≤ 2 log HIV-1 p24 pg/mL with 270 J/cm^2^ light dose, suggesting that 270 J/cm^2^ is the optimal light dose for HIV-1 inactivation in plasma under the conditions tested (Figure 2B). Further, we observed no significant increase in virus titers between day 14 and day 21 (*p* > 0.05). 

Based on the above observations, the remaining five different donor plasma samples were tested to validate viral inactivation by light treatment using only two light doses (27 J/cm^2^ and 270 J/cm^2^) and HIV-1 infectivity of the treated plasma was measured in H9 cells for up to 14 days only (Day 0, day 7 and day 14). As expected, all control plasma samples (Figure 3, D1–D5, D = donor) showed significant (*p* < 0.001) increases in virus titer close to 6 log HIV-1 p24 pg/mL with no observed statistically significant (*p* > 0.05) donor-to-donor variation. In case of the test plasma samples, while a 405 nm light dose of 27 J/cm^2^ failed to inactivate HIV-1 in the plasma of all five donors (Figure 3, 30 min, D1–D5), a higher dose of 270 J/cm^2^ consistently inactivated HIV-1 in all five donor plasma samples in the cell infectivity assay with no increase in HIV-1 titers (infectivity) at day 14 compared to control or low dose (27 J/cm^2^) light exposure (Figure 3, 5 h, D1–D5). These findings clearly validate that a 405 nm light dose of 270 J/cm^2^ is optimal for HIV-1 inactivation in human plasma under the conditions that were used in this study.

## 4. Discussion

The results reported here on evaluating the virucidal potential of 405 nm light clearly demonstrated that HIV-1 present in human plasma can be successfully inactivated by subjecting the plasma to an effective dose of 405 nm light (270 J/cm^2^). To our knowledge, this is the first report of 405 nm light-based inactivation of HIV-1 in human plasma. In line with the observations reported here, there have been reports of lipid-enveloped SARS-CoV-2 and Influenza A virus inactivation by 405 nm light whilst suspended in phosphate buffered saline (PBS) [13] and beta-coronaviruses (bovine) suspended in PBS as well as the virus suspension dried on steel surfaces [14]. Visible violet-blue light other than 405 nm wavelengths such as doses of 425 nm blue light were also reported to inhibit infection and replication of cell-associated SARS-CoV-2 by >99% 24 h post-infection and cell-free beta-coronaviruses including SARS-CoV-1, MERS-CoV, and SARS-CoV-2 up to 99.99% in a dose-dependent manner [15]. In a clinical study with another enveloped virus, hepatitis B virus (HBV), it was demonstrated that 2 h treatment of blood samples of HBV patients with 400–500 nm wavelength visible light in combination with riboflavin resulted in significant decrease of HBV DNA copy number in the blood [16]. A summary of virucidal effects of a range of visible wavelengths of blue light (401–470 nm) on a number of enveloped and non-enveloped viruses have been recently reviewed [17], with findings providing sufficient evidence that pathogen reduction technologies can incorporate safer visible violet-blue light in lieu of UV light for viral inactivation, in addition to its already well-established bacterial inactivation potential in various public health scenarios [18]. 

The mechanisms underlying the microbicidal effects of 405 nm light treatment on cellular microbes such as bacteria are well established [19]. The light treatment leads to the excitation of porphyrins and flavins present either within microbial cells or in surrounding biological milieu such as human blood and plasma (as in this study); absorption of 405 nm light by porphyrins and flavins results in the production of reactive oxygen species (ROS), including H_2_O_2_, OH or singlet oxygen, leading to oxidative damage and, ultimately, microbial cell death. More precisely, in cellular microbes, intracellular reactive oxygen species attack DNA, proteins, and membranes and, if the produced damage becomes too great, the microbial cell dies [20]. Human cells also contain such photosensitizers, but they have nevertheless proven to be resilient to visible light at the doses that kill pathogens, likely due to their more complex and effective oxidative stress response mechanisms [20]. Whilst viruses that do not have a cellular structure are also susceptible to the ROS induced effects produced by 405 nm light, the exact mechanisms of this damage on viral particles are not yet well understood.

We used high titer HIV-1 (10 ng/mL HIV-1 p24) spiked-plasma in our virus inactivation studies to evaluate the potential of violet-blue light-based technology and consistently demonstrated approximately 4–5 log reductions of HIV-1, which is comparable to the HIV-1 reductions reported in non-leuco-reduced platelet-rich plasma-derived platelets suspended in plasma using the Mirasol PRT System (≥4.19 log reduction) and the Intercept Blood System (≥4.23 log reduction) [5]. Further, we used a single strain of the highly virulent HIV-1 clade B subtype which is the dominant strain in North America. However, we believe the inactivation potential of 405 nm light on different HIV-1 variants will be similar, since the microbicidal effects of 405 nm light are through ROS generation-associated events. It is worth noting that treatment with a 270 J/cm^2^ dose of violet-blue light provided maximum viral inactivation, while with lower light doses the degree of viral inactivation in the plasma samples varied from donor to donor, although these results were not statistically significant. As HIV-1 infection of target cells is a multistep process starting with attachment of viral glycoproteins (gp120/gp41) to host cell receptors (CD4 and CCR5 or CXCR4), fusion of viral envelope, reverse transcription, integration, and maturation of virions by killing target cells [21], it will be interesting to understand which of these steps are affected by treatment with 405 nm light, and this will be the focus of future studies.

Our results demonstrate that inactivation of HIV-1 in human plasma is achievable with violet-blue light treatment. This observation, together with our previously demonstrated proof-of-concept data on the effectiveness of pathogen inactivation by 405 nm light, supports the potential utility of 405 nm light as a broad-spectrum pathogen reduction technology for further evaluation towards the safety of stored plasma and platelets preserved in plasma for transfusion.

## 5. Conclusions

The results obtained in this report demonstrate that HIV-1 can be successfully inactivated in plasma using 405 nm light at a dose of 270 J/cm^2^ under the conditions reported here. This study on HIV-1 opens other possibilities for future evaluation of the potential of 405 nm light for inactivation of other important blood-borne viruses such as hepatitis and arboviruses relevant to transfusion safety.

## Figures and Tables

**Figure 1 pathogens-11-00778-f001:**
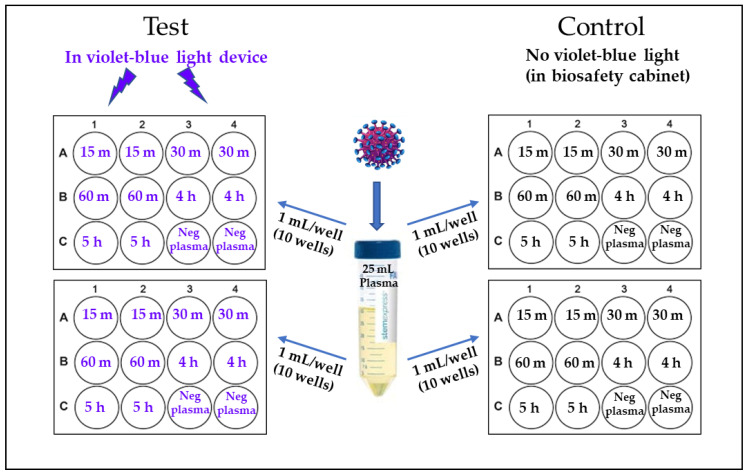
**Virus inactivation study design**. 25 mL of human plasma was spiked with HIV-1 clade B (final conc. 10 ng/mL of p24) and 1 mL/well was transferred to 10 wells in each plate. Remaining two wells in each plate were filled with virus-negative plasma. One set of two plates marked “Test” were treated with 405 nm violet-blue light and another set of two plates marked “Control” were placed in a biosafety cabinet with no violet-blue light. At each time point (15 min to 5 h), 1 mL samples from each well of the test and control plates were transferred to storage vials and stored at −80 °C, until further use.

**Figure 2 pathogens-11-00778-f002:**
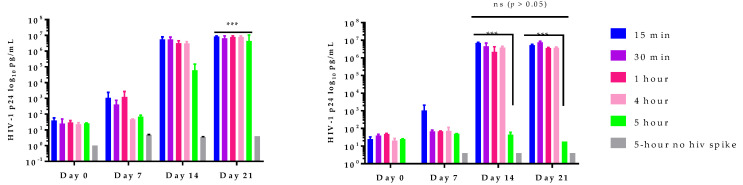
**405 nm light inactivates HIV-1.** HIV-1 spiked plasma samples were treated with violet-blue light (test) or not treated with violet-blue light (control). Both test and control plasma samples were periodically (15 m, 30 m, 1 h, 4 h and 5 h) removed from the 12-well plates and stored at −80 °C until infectivity study. Both plasma samples were used to infect HIV-1 permissive H9 cells. Culture supernatants were collected on first day of infection (day 0) to 21 days (day 21) of post infection. (**A**) HIV-1 spiked plasma without violet-blue light treatment (control) incubated in a biosafety cabinet for 15 min to 5 h, all have significant viral infection (~7 log HIV-1 p24) by day 21 compared to day 0 (*p* < 0.001). (**B**) HIV-1 spiked plasma treated with violet-blue light (test) for up to 5 h, exhibited significant (*p* < 0.001) reduction in the virus infectivity. Note that the light treatment for up to 4 h did not inactivate HIV-1 and no significant (ns) differences in HIV-1 p24 levels was observed between 14 day or 21 days post infection. Negative controls (without HIV-1 spike) of both for test and control were included to monitor cross contamination. HIV-1 p24 values are log transformed and plotted (*n* = 2 ± SD). Statistical analysis was executed using a two-way ANOVA wherein *** is *p* < 0.001 when compared to the intervals of light exposure. (ns) represents no significant differences observed in viral titers.

**Figure 3 pathogens-11-00778-f003:**
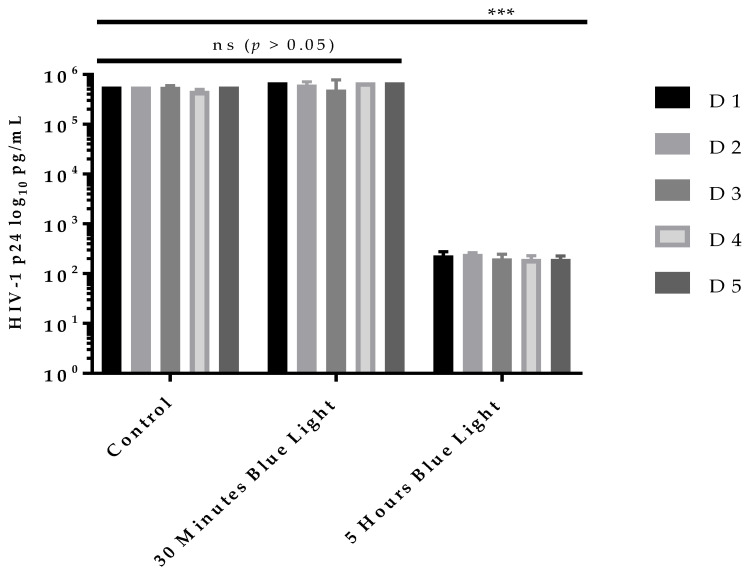
**405 nm light Inactivates HIV-1 in multiple plasma donors.** HIV-1 spiked plasma samples were treated with violet-blue light (test) or not treated with violet-blue light (control). Both test and control plasma samples were incubated for 30 min or 5 h and removed from the 12-well plates and stored at −80 °C until infectivity study. Both plasma samples were used to infect HIV-1 permissive H9 cells. Culture supernatants were collected up to 14 days of post infection. All 5 plasma control samples (D1C to D5C) spiked with HIV-1 and incubated in a biosafety cabinet for 30 min or 5 h have high HIV-1 titer (~6 log HIV-1 p24) at 14 days post infection. All 5 plasma test samples (D1T to D5T) exposed to violet-blue light for 30 min have high virus titer (~6 log HIV-1 p24) with no significant (ns, *p* > 0.05) difference to control HIV-1 viral titers. However, plasma samples light-treated for 5 h all have significant (*p* < 0.001) reduction in infectivity HIV-1 (~2 log HIV-1 p24) compared to controls or 30 min light treatment. HIV-1 p24 values are log transformed and plotted (*n* = 2 ± SD). Statistical analysis was executed using a two-way ANOVA wherein *** is *p* < 0.001 when compared to the control or 30 min of light exposure.

## Data Availability

The original contributions presented in the study are included in the article. All inquiries can be directed to the corresponding authors.

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
