# Peer review of "Visible 405 nm Violet-Blue Light Successfully Inactivates HIV-1 in Human Plasma"

_pathogens, 2022, doi:10.3390/pathogens11070778_

Round 1
Reviewer 1 Report
Contamination of blood products is generally a worldwide problem. The application of UVC radiation, which is otherwise often effective, is usually not possible here because blood products have a high UV absorption and thus very low penetration depth for the radiation. There are also approaches to disinfection with UVA radiation, which penetrates deeper into the plasma but requires additional photosensitizers.
The authors here report successful reduction of HIV in human plasma using visible violet light of 405 nm and without the addition of photosensitizers. The advantage is low exposure and potential degradation of plasma quality. The topic is highly interesting and fits into the current scientific discussion on the effect of visible light on viruses.
However, there are a few unclear points, especially with irradiation.
Major issues:
· I´m unsure about the illumination setup. If you have just a small light source above your shakers, the illumination might be less homogenous as you expect, e.g. due to shadowing effects of the tube walls. You should be aware of the fact, that the tube walls might look transparent but exhibit a very low transmission at 405 nm and these properties might even vary, if you have tubes of different manufacturers.
Did you measure the transmission? (You will hardly detect this with your eyes, because the 405 nm eye sensitivity is only 2% of the sensitivity at 550 nm.)
· How did you measure your intensity / dose?
· Please provide a transmission or absorption spectrum of your typical plasma. If the tube in Fig. 1 is real, I would guess that you won´t have any 405 nm irradiation left in the lower tube regions. My guess is based on the yellow impressions, which hints to a strong blue absorption and the very long path length of several centimeter.
It may be better to repeat the experiments with lower volumes and different vessels? (Glass or quartz beakers?)
· In addition, just for completeness: How does your shaker look like? Does it have a white or metallic surface that reflects your radiation? Have you taken this into account?
· From the presented data, I do not reach the conclusion why 270 J/cm2 should be the optimum? Maybe 500 J/cm2 would have been better but you did not try?
· Can you give an average reduction for the different irradiation doses?
· Can you explain, why the differences between the different doses in Fig. 2 seem to be so small?
·
Minor issues:
· Please enlarge Fig. 2. I have trouble reading it.
Could you please add the meaning of the left and right part of the figure?
Where do I find the control (no illumination) in the figure? (There is no hint in the legend?)
· Do you know, whether photosensitizers are expected to be in blood, naturally?
· What are UV light associates toxicities?
· What is a “saturation light dose”?
· You give your irradiation dose in your abstract, but no information of the achieved reduction. Please add the achieved reduction.
· References: Font size and font look strange?
Author Response
Please refer to attached document author cover letter 20237717v1.

Reviewer 2 Report
The authors have considered the relevant literature in this area, and have quoted this accordingly. Again nothing really to add there.
The study is well designed to address the question raised, and use appropriate methods for this work. The results are clearly and competently presented.
I have suggested a single amendment to Figure 2. (In Figure 2, it would be helpful to include the letters A. and B., which are referred to in the Figure legend)
The discussion of the results considers the results obtained, and presents these in the context of additional work addressing the antiviral actions of this light source.
Author Response
Reviewer 2
Comments and Suggestions for Authors
1. The authors have considered the relevant literature in this area and have quoted this accordingly. Again, nothing really to add there.
2. The study is well designed to address the question raised and use appropriate methods for this work. The results are clearly and competently presented.
3. I have suggested a single amendment to Figure 2. (In Figure 2, it would be helpful to include the letters A. and B., which are referred to in the Figure legend)
Authors’ response: As suggested, letters A and B were added to Fig. 2 in the revised version.
4. The discussion of the results considers the results obtained and presents these in the context of additional work addressing the antiviral actions of this light source.
Authors thank this reviewer for the supportive comments.